# STABILIZED SELF-TRAINING WITH NEGATIVE SAMPLING ON FEW-LABELED GRAPH DATA

## ABSTRACT

Graph neural networks (GNNs) are designed for semi-supervised node classification on graphs where only a small subset of nodes have class labels. However, under extreme cases when very few labels are available (*e.g.*, 1 labeled node per class), GNNs suffer from severe result quality degradation.

Specifically, we observe that existing GNNs suffer from *unstable* training process on few-labeled graph data, resulting to inferior performance on node classification. Therefore, we propose an effective framework, *Stabilized self-training with Negative sampling (SN)*, which is applicable to existing GNNs to stabilize the training process and enhance the training data, and consequently, boost classification accuracy on graphs with few labeled data. In experiments, we apply our SN framework to two existing GNN base models (GCN and DAGNN) to get SNGCN and SNDAGNN, and evaluate the two methods against 13 existing solutions over 4 benchmarking datasets. Extensive experiments show that the proposed SN framework is highly effective compared with existing solutions, especially under settings with very few labeled data. In particular, on a benchmark dataset Cora with only 1 labeled node per class, while GCN only has $44.6\%$ accuracy, SNGCN achieves $62.5\%$ accuracy, improving GCN by $17.9\%$; SNDAGNN has accuracy $66.4\%$, improving that of the base model DAGNN ($59.8\%$) by $6.6\%$.

## 1 INTRODUCTION

Graph is an expressive data model, representing objects and the relationships between objects as nodes and edges respectively. Graph data are ubiquitous with a wide range of real-world applications, *e.g.*, social network analysis (Qiu et al., 2018; Li & Goldwasser, 2019), traffic network prediction (Guo et al., 2019; Li et al., 2019), protein interface prediction (Fout et al., 2017), recommendation systems (Fan et al., 2019; Yang et al., 2020a). Among these applications, an important task is to classify the nodes in a graph into various classes. However, one tough situation commonly existing is the lack of sufficient labeled data, which are also expensive to collect.

To ease the situation, semi-supervised node classification on graphs has attracted much attention from both industry (Qiu et al., 2018; Li & Goldwasser, 2019) and academia (Defferrard et al., 2016; Hamilton et al., 2017; Velickovic et al., 2018; Liu et al., 2020; Li et al., 2018; Klicpera et al., 2019). It aims to leverage a small amount of labeled nodes and additionally a large amount of unlabeled nodes in a graph to train an accurate classifier. There exists a collection of graph neural networks for semi-supervised node classification (Kipf & Welling, 2017; Velickovic et al., 2018; Monti et al., 2017; Hamilton et al., 2017; Klicpera et al., 2019; Liu et al., 2020). For instance, Graph convolution networks (GCNs) rely on a message passing scheme called graph convolution that aggregates the neighborhood information of a node, including node features and graph topology, to learn node representations, which can then be used in downstream classification tasks (Kipf & Welling, 2017).

Despite the great success of GCNs, under the extreme cases when very few labels are given (*e.g.*, only one labeled node per class), the shallow GCN architecture, typically with two layers (Kipf & Welling, 2017), cannot effectively propagate the training labels over the input graph, leading to inferior performance. In particular, as shown in our experiments, on a benchmark dataset Cora with 1 labeled node per class (Cora-1), GCN is even less accurate than some unsupervised methods, such as DGI (Velickovic et al., 2019) and G2G (Bojchevski & Günnemann, 2018). Recently, several latest studies try to improve classification accuracy by designing deeper GNN architectures, *e.g.*, DAGNN

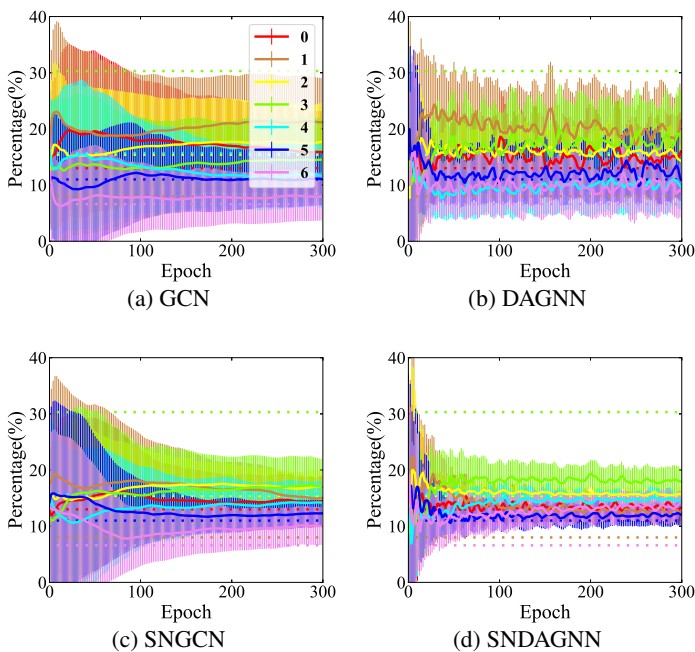

**Figure 1: The distribution of predicted labels in different classes in Cora-1.**

(Liu et al., 2020), which also address the over-smoothing issue identified in (Xu et al., 2018; Li et al., 2018; Chen et al., 2020a). However, these deep GNNs are still not directly designed to tackle the scarcity of labeled data, especially when only very few labels are available.

After conducting an in-depth study, we have an important finding that existing GNNs suffer from *unstable* training process, when labeled nodes are few. In particular, on Cora dataset with 7 classes, for each run, we randomly select 1 labeled node per class as the training data for both GCN and DAGNN, and repeat 100 runs with 300 epochs per run, to get the average number of predicted labels in percentage per class at each epoch and also the standard deviation. The statistical results of GCN and DAGNN are shown in Figures 1(a) and 1(b) respectively. $x$-axis is the epoch from 0 to 300, and $y$-axis is the percentage of a class in the predicted node labels. There are 7 colored lines representing the average percentage of the predicted labels of the respective classes, as the epoch increases. The dashed lines are the ground-truth percentage of each class in the Cora dataset. The shaded areas in colors represent the standard deviation. Observe that in Figure 1(a), GCN has high variance at different runs when predicting node labels, and the variance keeps large at late epochs, *e.g.*, 300, which indicates that GCN is quite unstable at different runs with 1 training label per class sampled randomly, leading to inferior classification accuracy as illustrated in our experiments. Moreover, as shown in Figure 1(b), DAGNN also suffers from unstable training process. The variance of DAGNN is relatively smaller than that of GCN, which provides a hint about why DAGNN performs better than GCN. Nevertheless, both GCN and DAGNN yield unstable training process with large variance. Since there is only 1 labeled node per class in Cora-1, at different runs, the randomly sampled training nodes can heavily influence the message passing process in GCN and DAGNN, depending on the connectivity of the training nodes to their neighborhoods over the graph topology, which result to the unstable training process observed above.

To address the unstable training process of existing GNNs when only very few labeled data are available, we propose a framework, *Stabilized self-training with Negative sampling* (SN), which is readily applicable to existing GNNs to improve classification accuracy via stabilized training process. In the proposed SN framework, at each epoch, we select a set of nodes with predicted labels of high confidence as pseudo labels and add such pseudo labels into training data to enhance the training of next epoch. To tackle the unstable issue of existing GNNs, we develop a stabilizing technique in self-training to balance the training. We then design a negative sampling regularization technique over pseudo labels to further improve node classification accuracy. In experiments, we apply our SN framework to GCN and DAGNN, denoted as SNGCN and SNDAGNN respectively.

Figures 1(c) and 1(d) report the average percentage and standard variance of the predicted labels per class per epoch of SNGCN and SNDAGNN on Cora-1 respectively. With our stabilized self-training technique, obviously, the variance of SNGCN in Figure 1(c) decreases quickly and becomes stable as epoch increases, compared with Figure 1(a) of GCN. SNDAGNN is also more stable than DAGNN as shown in Figures 1(d) and 1(b) respectively. As reported later in experiments, with the proposed SN framework, SNGCN achieves $62.5\%$ node classification accuracy on Cora-1, significantly improving GCN ($44.6\%$) by $17.9\%$, and SNDAGNN obtains $66.4\%$ accuracy on Cora-1 and outperforms DAGNN ($59.8\%$) by a substantial margin of $6.6\%$. We conduct extensive experiments on 4 benchmarking datasets, and compare with 13 existing solutions, to evaluate the performance of the proposed SN framework. Experimental results demonstrate that our SN framework is able to significantly improve classification accuracy of existing GNNs when only few labels are available, and is also effective when training labels are sufficient.

## 2 RELATED WORK

In literature, there are two directions to address the scarcity of labeled data for semi-supervised node classification: (i) explore multi-hop graph topological features to propagate the labels in $\mathcal{L}$ over the input graph, e.g., GCN (Kipf & Welling, 2017) and DAGNN (Liu et al., 2020); (ii) enhance the training data by pseudo labels (self-training) (Li et al., 2018) or augmenting graph data by new edges and features (Kong et al., 2020; Zhao et al., 2021). Note that these two directions are not mutually exclusive, but can work together on few-labeled graph data. Here we review the existing studies that are most relevant to this paper.

**GNNs.** There exist a large collection of GNNs, such as GCN, DAGNN, GAT, MoNet, and APPNP (Kipf & Welling, 2017; Liu et al., 2020; Bruna et al., 2014; Henaff et al., 2015; Defferrard et al., 2016; Velickovic et al., 2018; Monti et al., 2017; Chen et al., 2020b; Klicpera et al., 2019). We introduce the details of GCN (Kipf & Welling, 2017) and DAGNN (Liu et al., 2020) here. GCN learns the representation of each node by iteratively aggregating the representations of its neighbors. Specifically, GCN consists of $k > 0$ layers, each with the same propagation rule defined as follows. At the $\ell$-th layer, the representations $\mathbf{H}^{(\ell-1)}$ of previous layer are aggregated to get $\mathbf{H}^{(\ell)}$.

$$\mathbf{H}^{(\ell)} = \sigma(\hat{\mathbf{A}}\mathbf{H}^{(\ell-1)}\mathbf{W}^{(\ell)}), \ell = 1, 2, ..., k. \tag{1}$$

$\hat{\mathbf{A}} = \tilde{\mathbf{D}}^{-\frac{1}{2}}\tilde{\mathbf{A}}\tilde{\mathbf{D}}^{-\frac{1}{2}}$ is the graph laplacian, where $\tilde{\mathbf{A}} = \mathbf{A} + \mathbf{I}$ is the adjacency matrix of $\mathcal{G}$ after adding self-loops ($\mathbf{I}$ is the identity matrix) and $\tilde{\mathbf{D}}$ is a diagonal matrix with $\tilde{\mathbf{D}}_{ii} = \sum_j \tilde{\mathbf{A}}_{ij}$. $\mathbf{W}^{(\ell)}$ is a trainable weight matrix of the $\ell$-th layer, and $\sigma$ is a nonlinear activation function. Initially, $\mathbf{H}^{(0)} = \mathbf{X}$. Note that GCN usually achieves superior performance with 1-layer or 2-layer models (Kipf & Welling, 2017). When applying multiple layers to leverage large receptive fields, the performance degrades severely, due to the over-smoothing issue identified in (Xu et al., 2018; Li et al., 2018; Chen et al., 2020a). A recent deep GNN architecture, DAGNN, tackles the over-smoothing issue and achieves state-of-the-art results by decoupling representation transformation and propagation in GNNs (Liu et al., 2020). Then it utilizes an adaptive adjustment mechanism to balance the information from local and global neighborhoods of each node. Specifically, the mathematical expression of DAGNN is as follows. DAGNN uses a learnable parameter $\mathbf{s} \in \mathbb{R}^{c \times 1}$ to adjust the weight of embeddings at different propagation level (from 1 to $k$). It processes data in the following way. $\mathbf{Z} = \text{MLP}(\mathbf{X}) \in \mathbb{R}^{n \times c}$, $\mathbf{H}_\ell = \hat{\mathbf{A}}^\ell \cdot \mathbf{Z} \in \mathbb{R}^{n \times c}, \ell = 1, 2, ..., k$, $\mathbf{S}_\ell = \mathbf{H}_\ell \cdot \mathbf{s} \in \mathbb{R}^{n \times 1}, \ell = 1, 2, ..., k$, $\hat{\mathbf{S}}_\ell = [\mathbf{S}_\ell, \mathbf{S}_\ell, ..., \mathbf{S}_\ell] \in \mathbb{R}^{n \times c}, \ell = 1, 2, ..., k$, $\mathbf{X}_{out} = \text{softmax}(\sum_{\ell=1}^k \mathbf{H}_\ell \odot \hat{\mathbf{S}}_\ell)$, where $\hat{\mathbf{A}}^\ell$ is the $\ell$-th power of matrix $\hat{\mathbf{A}}$, $\odot$ is the Hadamard product, $\cdot$ is dot product, MLP is the Multilayer Perceptron and softmax operation is on the second dimension.

**Data Augmentation.** Another way to address the situation of limited labeled data is to add pseudo labels to training dataset by self-training (Li et al., 2018), or enhance the graph data by adding new edges and features (Zhao et al., 2021; Kong et al., 2020). Self-training itself is a general methodology (Scudder, 1965) and is used in various domains in addition. It is used in word-sense disambiguation (Yarowsky, 1995; Hearst, 1991), bootstrap for information extraction and learning subjective nouns (Riloff & Jones, 1999), and text classification (Nigam et al., 2000). In (Zhou et al., 2012), it suggests that selecting informative unlabeled data using a guided search algorithm can significantly improve performance over standard self-training framework. Buchnik & Cohen (2018) mainly consider self-training for diffusion-based techniques. Recently, self-training has been adopted for semi-supervised

tasks on graphs. For instance, Li et al. (2018) propose self-training and co-training techniques for GCN. This self-training work selects the top-$k$ confident predicted labels as pseudo labels. The co-training technique co-trains a GCN with a random walk model to handle few-labeled data. Compared with existing self-training work, our framework are different as shown later. In particular, our framework has a different strategy to select pseudo labels and also has a stabilizer to address the deficiencies of existing GNNs; moreover, we propose a negative sampling regularization technique to further boost accuracy. Besides, in existing work, if a node is selected as a pseudo label, it will never be moved out even if the pseudo label becomes obviously wrong in later epochs. On the other hand, in our framework, we update pseudo labels in each epoch to avoid such an issue. There also exist studies to augment the original graph data, which is different from self-training. For instance, Zhao et al. (2021) utilize link prediction to promote intra-class edges and demote inter-class edges in a given graph. Kong et al. (2020) iteratively augment node features with gradient-based adversarial perturbations to enhance the performance of GNNs.

## 3 THE FRAMEWORK

### 3.1 PROBLEM FORMULATION

Let $\mathcal{G} = (\mathcal{V}, \mathcal{E}, \mathbf{X})$ be a graph consisting of a node set $\mathcal{V}$ with cardinality $n$, a set of edges $\mathcal{E}$ of size $m$, each connecting two nodes in $\mathcal{V}$, a feature matrix $\mathbf{X} \in \mathbb{R}^{n \times d}$, where $d$ is the number of features in $\mathcal{G}$. For every node $v_i \in \mathcal{V}$, it has a feature vector $\mathbf{X}_i \in \mathbb{R}^d$, where $\mathbf{X}_i$ is the $i$-th row of $\mathbf{X}$. Let $c$ be the number of classes in $\mathcal{G}$. We use $\mathcal{L}$ to denote the set of labeled nodes, and obviously $\mathcal{L} \subseteq \mathcal{V}$. Let $\mathcal{U}$ be the set of unlabeled nodes and $\mathcal{U} = \mathcal{V} \setminus \mathcal{L}$. Each labeled node $v_i \in \mathcal{L}$ has a one-hot vector $\mathbf{Y}_i \in \{0, 1\}^c$, indicating the class label of $v_i$. Under the few-labeled setting, $|\mathcal{L}| \ll |\mathcal{U}|$. A high-level definition of the semi-supervised node classification problem is as follows.

**Definition 1.** *Given a graph* $\mathcal{G} = (\mathcal{V}, \mathcal{E}, \mathbf{X})$*, a set of labeled nodes* $\mathcal{L} \subseteq \mathcal{V}$*, and a groundtruth class label* $\mathbf{Y}_i \in \{0, 1\}^c$ *per node* $v_i \in \mathcal{L}$*, assuming that each node belongs to exactly one class, Semi-Supervised Node Classification predicts the labels of the unlabeled nodes.*

In particular, the aim is to leverage the graph $\mathcal{G}$ with the labeled nodes in $\mathcal{L}$, and to train a forward predicting classification model/function $f(\mathcal{G}, \theta)$ that takes as input the graph $\mathcal{G}$ and a set of trainable parameters $\theta$. The output of $f$ is a matrix $\mathbf{F} \in \mathbb{R}^{n \times c}$, with each $i$-th row $\mathbf{F}_i \in [0, 1]^c$ representing the output probability vector of node $v_i \in \mathcal{V}$ (the 1-norm of $\mathbf{F}_i$ is normalized to 1).

We adopt the widely used cross-entropy loss. For a node $v_i$, its loss of $\mathbf{F}_i$ with respect to its true class label $\mathbf{Y}_i$, $L(\mathbf{Y}_i, \mathbf{F}_i)$, is defined as follows.

$$L(\mathbf{Y}_i, \mathbf{F}_i) = -\sum_{j=1}^{c} \mathbf{Y}_{i,j} \ln(\mathbf{F}_{i,j})$$

where $\mathbf{Y}_{i,j}$ is the $j$-th value in $\mathbf{Y}_i$ and $\mathbf{F}_{i,j}$ is the $j$-th value in $\mathbf{F}_i$.

### 3.2 STABILIZED SELF-TRAINING

Recall that existing GNNs suffer from unstable training process as shown in Figure 1. In this section, we present the stabilized self-training technique that not only augments training data with pseudo labels but also stabilizes the training process. We first explain how to choose pseudo labels and then introduce loss function of stabilized self-training.

At a certain epoch, given the matrix $\mathbf{F} \in \mathbb{R}^{n \times c}$, with each $i$-th row $\mathbf{F}_i \in [0, 1]^c$ representing the output probability vector of node $v_i \in \mathcal{V}$. Let $\tilde{\mathbf{Y}}_{i,j}$ satisfying the following Eq. (2) be the predicted label of node $v_i$. We say that, with confidence $\mathbf{F}_{i,j}$, node $v_i$ has class label $\mathcal{C}_j$; *i.e.*, the largest element $\mathbf{F}_{i,j}$ in vector $\mathbf{F}_i$ is called the confidence of node $v_i$.

$$\tilde{\mathbf{Y}}_{i,j} = \begin{cases} 1 & \text{if } j = \arg\max_{j'} \mathbf{F}_{i,j'}, \\ 0 & \text{otherwise,} \end{cases} \tag{2}$$

Then for every unlabeled node $v_i \in \mathcal{U}$, we can get $N_i$, the number of nodes with the same predicted label as $v_i$, in Eq. (3). Recall that in Figure 1, in existing GCN and DAGNN, the distribution of

the predicted labels is unstable with large variance during the training process; we also observe that most predicted labels are in the same class, especially at the early epochs. To reduce such unstable situation of existing GNNs, we use $N_i$ as a stabilizer in our framework to be explained shortly.

$$N_i = \left| \left\{ v_j \in \mathcal{U} \middle| \tilde{\mathbf{Y}}_i = \tilde{\mathbf{Y}}_j \right\} \right| \tag{3}$$

For every unlabeled nodes $v_i$ in $\mathcal{U}$, it can have a predicted label. However, the confidence $\arg\max_{j'} \mathbf{F}_{i,j'}$ might be low. We do not want to add such low-confidence labels into the training of the next epoch. Therefore, we only choose those unlabeled nodes with high-confidence predicted labels as pseudo labels to be augmented into the training data of next epoch. In particular, an unlabeled node $v_i$ is selected to be a node with *pseudo label* in next epoch, if its confidence satisfies a threshold $\beta$, as shown below. We use $\mathcal{U}'$ to denote all unlabeled nodes selected with pseudo labels.

$$\mathcal{U}' = \left\{ v_i \in \mathcal{U} \middle| \max_j \mathbf{F}_{i,j} > \beta \right\} \tag{4}$$

where $\beta \in [0, 1]$ is a threshold controlling the extent of cautious selection for self-training. A bigger threshold means stricter selection of the pseudo labels.

After explaining how to choose pseudo labels above, we present the loss of our stabilized self-training technique in Eq. (5). In particular, we design $\frac{1}{N_i+1}$ as the stabilizer of the training process, to overcome the deficiencies of existing GNNs illustrated in Figure 1. The intuition is that, if an unlabeled node $v_i$ is in a pseudo label class with many nodes, its importance in the loss function is reduced. In other words, our stabilized self-training loss reduces the impact of classes with many pseudo labeled nodes, which is especially useful to rectify the training process when the predictions in the early epochs are incorrect or less confident, compared with ground truth.

$$L_{sst} = \sum_{\forall v_i \in \mathcal{U}'} \frac{1}{N_i + 1} \cdot L(\tilde{\mathbf{Y}}_i, \mathbf{F}_i) \tag{5}$$

Compared with existing self-training techniques (Li et al., 2018), our stabilized self-training technique has major differences. First, we develop the stabilizer to re-weight the importance of pseudo labels in the loss function, so as to address the unstable issue of existing GNNs. Second, we select only those nodes with high-confidence pseudo labels satisfying $\beta$ threshold, and adaptively update the pseudo labels *per epoch*, meaning that a pseudo label in previous epoch will be removed in the next epoch if its confidence becomes low. On the other hand, existing methods keep a pseudo label once it is selected and never remove it in later epochs (Li et al., 2018), which may harm the training quality if the pseudo label is wrong compared with ground truth.

### 3.3 NEGATIVE SAMPLING REGULARIZATION

Under extreme cases with very few labeled nodes (*e.g.*, 1 labeled node per class), we further design a negative sampling regularization technique for better performance. In existing studies, negative sampling is used as an unsupervised technique over node embeddings in network embedding methods (Yang et al., 2020b; Velickovic et al., 2019; Yang et al., 2020b). Here we customize it to the semi-supervised node classification task, and apply negative sampling over *labels* instead of embeddings.

Intuitively, the label of a node $v$ should be distant to the label of another node $u$ if these two nodes are faraway on the input graph $\mathcal{G}$. Specifically, a positive sample is a node $v_i$ in $\mathcal{L}$ or $\mathcal{U}'$. We sample a set $\mathcal{I}$ of positive samples from $\mathcal{L} \cup \mathcal{U}'$ uniformly at random. The negative samples of a positive sample $v_i$ are the nodes that are not directly connected to $v_i$ in graph $\mathcal{G}$. For *each* positive sample $v_i$ in $\mathcal{I}$, we sample a fixed-size set $\mathcal{J}_i$ of negative samples uniformly at random.

For a positive-negative pair $(v_i, v_j)$, compared with the $\tilde{\mathbf{Y}}_i$ of $v_i \in \mathcal{L} \cup \mathcal{U}'$, the intention is to let the output vector $\mathbf{F}_j$ of $v_j$ to be as different as possible. Here we use the symbol $\tilde{\mathbf{Y}}_i$ to represent the pseudo label $\tilde{\mathbf{Y}}_i$ for node $v_i$ in $\mathcal{U}'$ or ground-truth label $\mathbf{Y}_i$ of node $v_i$ in $\mathcal{L}$ to avoid ambiguity. Denote $\mathbf{1}$ as the all-one vector in $\mathbb{R}^c$. Then we have the following loss of all positive-negative pairs.

$$L_{neg} = \sum_{\forall v_i \in \mathcal{I}} \sum_{\forall v_j \in \mathcal{J}_i} \frac{1}{|\mathcal{I}| \cdot |\mathcal{J}_i|} L(\tilde{\mathbf{Y}}_i, \mathbf{1} - \mathbf{F}_j) \tag{6}$$

---

**Algorithm 1:** SN Framework Over GNNs

---

1   **Input**: Graph $\mathcal{G} = (\mathcal{V}, \mathcal{E}, \mathbf{X})$ with labeled node set $\mathcal{L}$ and unlabeled node set $\mathcal{U}$
2   **Output**: the learned classifier $f(\cdot, \theta)$.
3   Generate initial parameter $\theta$ for model $f(\cdot, \cdot)$.
4   **for** each epoch $t = 0, 1, 2, ..., T$ **do**
5       Use GNN to compute prediction $\mathbf{F} \leftarrow f(\mathcal{G}, \theta)$
6       Get high confidence set $\mathcal{U}'$ and its stabilizing factor $\frac{1}{N_i+1}$ per node $v_i$(Section 3.2)
7       Get positive samples and corresponding negative samples using $\mathcal{L} \cup \mathcal{U}'$ and $\mathcal{G}$ (Section 3.3)
8       Get $L_{total}$ of current epoch by Eq. (7) (Section 3.4)
9       Update model parameters by $\theta \leftarrow$ Adam Optimizer$(\theta, gradient = \nabla_\theta L_{total})$.
10      **if** Convergence **then**
11         Break
12      **end**
13 **end**

---

### 3.4   Objective Function and Algorithm

Our final loss function is as follows, and it combines the stabilized self-training loss and negative sampling loss in Eq. (5) and Eq. (6) respectively.

$$L_{total} = \frac{1}{|\mathcal{L}|} \cdot \sum_{\forall v_i \in \mathcal{L}} L(\mathbf{Y}_i, \mathbf{F}_i) + \lambda_1 L_{sst} + \lambda_2 L_{neg}, \tag{7}$$

where $\lambda_1$ and $\lambda_2$ are factors controlling the impact of these two losses.

Algorithm 1 shows the pseudo-code of our SN framework over GNNs, and it takes as input a graph $\mathcal{G}$ with labeled nodes $\mathcal{L}$ and unlabeled nodes $\mathcal{U}$. Note that SN can be instantiated over either a shallow or a deep GNN, *e.g.*, GCN and DAGNN introduced in Section 2. The output of Algorithm 1 is the learned classification model $f$ with trainable parameters $\theta$. At Line 3, SN initializes the trainable parameters $\theta$ by Xavier (Glorot & Bengio, 2010). Then from Lines 4 to 12, SN trains the classification model per epoch $t$ iteratively, until convergence or the max number $T$ of iterations is reached. Specifically, at Line 5, SN first use a GNN to obtain the forward prediction output $\mathbf{F}$. Then at Line 6, SN detects the pseudo-labeled set $\mathcal{U}'$ and obtain the stabilizer $\frac{1}{N_i+1}$ of each node $v_i$ in $\mathcal{U}'$, after which, at Line 7 we perform negative sampling to obtain positive samples $\mathcal{I}$ and negative samples $\mathcal{J}_i$. At Line 8, SN computes loss $L_{total}$ of current epoch according to Eq. (7). And at Line 9, SN updates model parameters $\theta$ for next epoch by Adam optimizer (Kingma & Ba, 2015).

## 4   Experiments

We evaluate SN against 13 competitors for semi-supervised node classification on 4 benchmark graph datasets. All experiments are conducted on a machine powered by an Intel(R) Xeon(R) E5-2603 v4 @ 1.70GHz CPU, 131GB RAM, 16.04.1-Ubuntu, and 4 Nvidia Geforce 1080ti Cards with Cuda version 10.2. Source codes of all competitors are obtained from the respective authors. Our SN framework is implemented in Python, using libraries including PyTorch (Paszke et al., 2019) and PyTorch Geometric (Fey & Lenssen, 2019). An anonymous link[1] of our source code is provided.

### 4.1   Datasets and Competitors

**Datasets.** Table 1 shows the statistics of the 4 real-world graphs used in our experiments. We list the number of nodes, edges, features and classes in each graph dataset respectively. Specifically, the 4 datasets are Cora (Sen et al., 2008), Citeseer (Sen et al., 2008), Pubmed (Sen et al., 2008), and Core-full (Bojchevski & Günnemann, 2018), all of which are widely used for benchmarking node classification performance in existing studies (Sun et al., 2020; Li et al., 2018; Liu et al., 2020). Notice that every node in these graphs has a ground-truth class label.

---

[1]https://anonymous.4open.science/r/e7aca211-0d8d-4564-8f3f-0ef24b01941e/

**Table 1: Datasets**

|  | Cora | Citeseer | Pubmed | Cora-full |
|---|---|---|---|---|
| # of Nodes | 2708 | 3327 | 19717 | 19793 |
| # of Edges | 5429 | 4732 | 44338 | 65311 |
| # of Features | 1433 | 3703 | 500 | 8710 |
| # of Classes | 7 | 6 | 3 | 67 |

**Competitors.** We compare with 13 existing solutions, including LP (Label Propagation) (Wu et al., 2012), DeepWalk (Perozzi et al., 2014), LINE (Tang et al., 2015), G2G (Bojchevski & Günnemann, 2018), DGI (Velickovic et al., 2019), GCN (Kipf & Welling, 2017), GAT (Velickovic et al., 2018), MoNet (Monti et al., 2017), APPNP (Klicpera et al., 2019), DAGNN (Liu et al., 2020), STs (Li et al., 2018), LCGCN and LCGAT in (Xu et al., 2020). In particular, GCN, GAT, MoNet, APPNP, and DAGNN are GNNs. DeepWalk, DGI, LINE, and G2G are unsupervised network embedding methods. STs represents the four variants in (Li et al., 2018), including Self-Training, Co-Training, Union, and Intersection; we summarize the best results among them as the results of STs.

## 4.2 EXPERIMENTAL SETTINGS

We evaluate our framework and the competitors on semi-supervised node classification tasks with various settings. In particular, for each graph dataset, we repeat experiments on 100 random data splits as suggested in (Liu et al., 2020; Li et al., 2018) and report the average performance. For each graph dataset, we vary the number of labeled nodes per class in $\{1, 3, 5, 10, 20\}$, where $1, 3, 5$ represent the very few-labeled settings. Following convention in existing work (Liu et al., 2020), we explain what a random data split is, as follows. For example, when the number of labeled nodes per class on Cora is 3 (denoted as Cora-3), since Cora has 7 classes, we randomly pick 3 nodes per class, combining together as a training set of size 21 (*i.e.*, the labeled node set $\mathcal{L}$), and then, among the remaining nodes, we randomly select 500 nodes as a validation set, and 1000 nodes as a test set. Each data split consists of a training set, a validation set, and a test set as mentioned above. We use the classification accuracy on test set as evaluation metric. Specifically, accuracy is defined as the fraction of the testing nodes whose class labels are correctly predicted by the learned classifier.

## 4.3 IMPLEMENTATION DETAILS

We instantiate SN framework over the classic GCN model with 2 layers and a recent deep GNN architecture DAGNN to demonstrate the effectiveness and applicability of SN. The instantiation of SN over GCN and DAGNN are dubbed as SNGCN and SNDAGNN respectively. SNGCN and SNDAGNN has parameters (i) inherited from GCN and DAGNN and (ii) developed in SN. Hence, we first tune the best parameters of the base models under each classification task setting on each dataset and report this result for them for a fair comparison.

**Base models (GCN and DAGNN).** In base models, we tune the parameters: a $L_2$ regularization rate with search space in $\{1e\text{-}2, 5e\text{-}3, 1e\text{-}3, 5e\text{-}4, 1e\text{-}4, 5e\text{-}5, 0\}$, a dropout rate in $\{0.5, 0.8\}$. For DAGNN, the level $k$ of propagation after MLP is searched in $\{10, 15, 20\}$. We have the following parameters for base models in experiments: the number of hidden units of GCN and MLP (in DAGNN) is 64 units without bias; the number of layers of GCN and MLP (in DAGNN) is 2 layers; the learning rate of Adam Optimizer is 0.01; the activation function is RELU; the maximum number of training epochs is 1000. Moreover, early stopping is triggered when the validation loss is smaller than the average validation loss of previous 100 epochs, and the current epoch is beyond 500 epochs.

**SN over GCN and DAGNN (SNGCN and SNDAGNN).** After finding the best hyper parameters of the base models, we then tune the parameters in SN. $\lambda_1$ is searched in $\{0.1, 1\}$ and $\lambda_2$ is searched in $\{0, 0.1, 1\}$. Stabilizing enabler searches in {True, False}. The number of positive and negative samples ($|\mathcal{I}|, |\mathcal{J}_i|$) is searched in $\{(1, 10), (2, 5), (5, 2), (10, 1)\}$. For instance, $(2, 5)$ means that we sample 2 positive nodes and then for each positive node, we sample 5 negative nodes.

**Competitors.** We use the parameters suggested in the original papers of the competitors to tune their models, and report the best results of the competitors. Notice that for unsupervised network embedding methods, including DeepWalk, DGI, LINE, and G2G, after obtaining the embeddings, we use logistic regression to train a node classifier over the embedding (Velickovic et al., 2019).

**Table 2: Accuracy results (in percentage) on Cora and CiteSeer respectively, averaged over 100 random data splits.** (The **best** accuracy is in **bold**.)

| # of Labels per class | Cora | | | | | CiteSeer | | | | |
|---|---|---|---|---|---|---|---|---|---|---|
| | 1 | 3 | 5 | 10 | 20 | 1 | 3 | 5 | 10 | 20 |
| **GCN** | 44.6 | 63.8 | 71.3 | 77.2 | 81.4 | 40.4 | 53.5 | 61.0 | 65.8 | 69.5 |
| **SNGCN (ours)** | 62.5 | 72.8 | 75.8 | 80.7 | 82.5 | **56.2** | **66.4** | 68.0 | 70.2 | **72.1** |
| **DAGNN** | 59.8 | 72.4 | 76.7 | 80.8 | 83.7 | 46.5 | 58.8 | 63.6 | 67.9 | 71.2 |
| **SNDAGNN (ours)** | **66.4** | **77.6** | **79.8** | **82.2** | **84.1** | 48.5 | 65.9 | 67.9 | 69.8 | **72.1** |
| **LP** | 51.5 | 60.5 | 62.5 | 64.2 | 67.3 | 30.1 | 37.0 | 39.3 | 41.9 | 44.8 |
| **DeepWalk** | 40.4 | 53.8 | 59.4 | 65.4 | 69.9 | 28.3 | 34.7 | 38.1 | 42.0 | 45.6 |
| **LINE** | 49.4 | 62.6 | 63.4 | 71.1 | 74.0 | 28.0 | 34.7 | 38.0 | 43.1 | 48.5 |
| **G2G** | 54.5 | 68.1 | 70.9 | 73.8 | 75.8 | 45.1 | 56.4 | 60.3 | 63.1 | 65.7 |
| **DGI** | 55.3 | 70.9 | 72.6 | 76.4 | 77.9 | 46.1 | 59.2 | 64.1 | 67.6 | 68.7 |
| **STs** | 53.1 | 67.3 | 72.5 | 76.2 | 79.8 | 37.2 | 51.8 | 60.7 | 67.4 | 70.2 |
| **GAT** | 41.8 | 61.7 | 71.1 | 76.0 | 79.6 | 32.8 | 48.6 | 54.9 | 60.8 | 68.2 |
| **MoNet** | 43.4 | 61.2 | 70.9 | 76.1 | 79.3 | 38.8 | 52.9 | 59.7 | 64.6 | 66.9 |
| **APPNP** | 44.7 | 66.3 | 74.1 | 79.0 | 81.9 | 34.6 | 52.2 | 59.4 | 66.0 | 71.8 |
| **LCGCN** | 63.6 | 74.4 | 77.5 | 80.4 | 82.4 | 55.3 | 59.0 | **68.4** | 70.3 | **72.1** |
| **LCGAT** | 58.7 | 74.5 | 77.5 | 79.7 | 82.6 | 50.9 | 66.3 | 68.5 | **70.9** | 71.5 |

## 4.4 OVERALL RESULTS

Table 2 reports the classification accuracy (in percentage) of all methods on Cora and CiteSeer, when varying the number of labeled nodes per class in {1,3,5,10,20}. The first and second rows report the performance of GCN and SNGCN. The third and fourth rows report the performance of DAGNN and SNDAGNN. Observe that SNGCN (resp. SNDAGNN) enhanced by our SN framework significantly outperforms GCN (resp. DAGNN) under all task settings, and the performance gain of SNGCN is especially significant when the labels per class are few. For instance, on Cora-1, SNGCN has accuracy 62.5%, while the accuracy of GCN is 44.6%, indicating that the proposed framework improves GCN by 17.9%. This demonstrates the power of the proposed SN framework to boost classification performance. Compared with other competitors, observe that SNDAGNN has the best performance under all settings of Cora (in bold), and SNGCN has the best performance on CiteSeer-1 and 3 and 20, while achieving similar performance compared with LCGCN and LCGAT on CiteSeer-5 and 10. Apart from the superior performance of our SN framework over GCN and DAGNN, in Table 2, we can observe two interesting findings. First, under extremely-few-labels settings (e.g., Cora-1), unsupervised methods G2G (54.5%) and DGI (55.3%) achieve better performance than GCN (44.6%). One reason is that the unsupervised methods are good at cases when no labeled data are available, while GCNs still require a sufficient amount of labeled data. This finding demonstrates the intuition of the negative sampling regularization in Section 3.3, and also sheds light on possible future research to use unsupervised techniques to further enhance the performance of semi-supervised learning. Second, the performance gap between our methods and competitors enlarges as the number of labels per class decreases, which further illustrates the effectiveness of the proposed SN framework under extreme settings on graphs with very few-labeled nodes per class.

Table 3 presents the classification accuracy of all methods under all settings in {1,3,5,10,20} on PubMed and Cora-full datasets. We exclude from this table, the inaccurate competitors (*e.g.*, Deep-Walk and LINE) that are obviously outperformed by other competitors. Observe that on PubMed and Cora-full, SNDAGNN and SNGCN achieve the higher accuracy consistently than their respective base models GCN and DAGNN. For instance, on Cora-full-1, SNGCN achieves 30.8% accuracy, 6.3% better than GCN. Moreover, observe that on PubMed, SNDAGNN consistently outperforms all other competitors, *e.g.*, LCGCN; on Cora-full, SNGCN outperforms all competitors on 1 and 3 settings, and SNDAGNN has the best accuracy on 5, 10, and 20 settings.

In summary, the experimental results presented in Tables 2 and 3 validate the effectiveness of the proposed SN framework over GCN and DAGNN to boost classification performance, especially when only very few labeled nodes are available.

**Table 3: Accuracy results (in percentage) on PubMed and Cora-full respectively, averaged over 100 random data splits. (** The **best** accuracy is in **bold**.)

| # of Labels per class | PubMed | | | | | Cora-full | | | | |
|---|---|---|---|---|---|---|---|---|---|---|
| | 1 | 3 | 5 | 10 | 20 | 1 | 3 | 5 | 10 | 20 |
| **GCN** | 55.5 | 66.0 | 70.4 | 74.6 | 78.7 | 24.5 | 41.4 | 48.1 | 55.8 | 60.2 |
| **SNGCN (ours)** | 60.8 | 67.8 | 71.6 | 76.1 | 79.4 | **30.8** | **44.9** | 49.4 | 56.6 | 60.9 |
| **DAGNN** | 59.4 | 69.5 | 72.0 | 76.8 | 80.1 | 27.3 | 43.2 | 49.8 | 55.8 | 60.4 |
| **SNDAGNN (ours)** | **61.0** | **72.1** | **74.9** | **78.2** | **80.6** | 27.6 | 44.4 | **51.1** | **56.8** | **61.2** |
| **LP** | 55.7 | 61.9 | 63.5 | 65.2 | 66.4 | 26.3 | 32.4 | 35.1 | 38.0 | 41.0 |
| **G2G** | 55.2 | 64.5 | 67.4 | 72.0 | 74.3 | 25.8 | 36.4 | 43.3 | 49.3 | 54.3 |
| **DGI** | 55.1 | 63.4 | 65.3 | 71.8 | 73.9 | 26.2 | 37.9 | 46.5 | 55.3 | 59.8 |
| **ST$_S$** | 55.1 | 65.4 | 69.7 | 74.0 | 78.5 | 29.2 | 43.6 | 48.9 | 53.4 | 60.8 |
| **APPNP** | 54.8 | 66.9 | 70.8 | 76.0 | 79.4 | 24.3 | 41.5 | 48.5 | 55.3 | 60.1 |
| **GAT** | 52.7 | 64.4 | 69.4 | 73.7 | 73.5 | 24.8 | 41.0 | 47.5 | 54.7 | 59.9 |
| **LCGCN** | 56.6 | 69.2 | 72.6 | 74.6 | 80.0 | 26.7 | 43.9 | 49.2 | 55.9 | 60.5 |
| **LCGAT** | 49.5 | 59.2 | 62.3 | 70.2 | 65.3 | 27.4 | 43.2 | 48.4 | 55.0 | 60.1 |

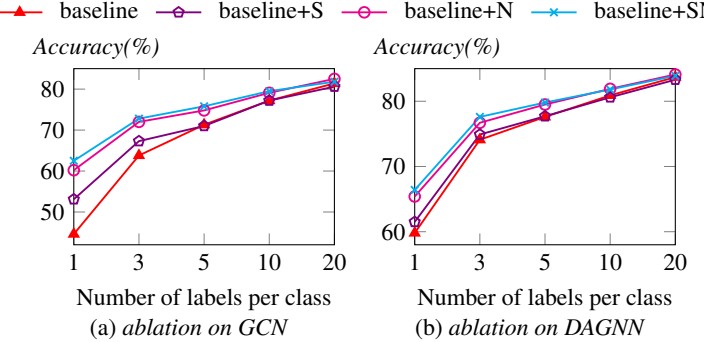

**Figure 2: Ablation study of SN on Cora.**

## 4.5 ABLATION STUDY

We conduct ablation study to evaluate the contributions of the techniques of SN presented in Section 3. Denote baseline+SN as the method with the whole SN framework enabled, baseline+S as the method with only stabilized self-training loss in Eq. (5) enabled, and baseline+N as the method with only negative sampling loss in Eq. (6) enabled. Figures 2a and 2b report the ablation results on baselines GCN and DAGNN respectively, on Cora when varying the number of labels per class in $\{1, 3, 5, 10, 20\}$. Observe that the accuracies of baseline+S and baseline+N are always better than the baseline model, *i.e.*, GCN and DAGNN. Also the accuracy of baseline+SN is almost always the highest under all settings. The ablation study demonstrates the power of our proposed techniques to improve classification accuracy.

## 5 CONCLUSION

This paper presents Stabilized self-training with Negative sampling (SN), an effective framework for semi-supervised node classification on few-labeled graph data. SN achieves superior performance on graphs with extremely few labeled nodes, through two main designs: a stabilized self-training technique that adaptively selects and re-weights high-confidence pseudo labels based on the confidence distribution of current epoch to enhance the training process, and a negative sampling regularization that further fully utilizes unlabeled data to train a high-quality classifier. The effectiveness of SN is extensively evaluated on 4 real graphs, compared against 13 existing solutions. Regarding future work, we plan to enhance SN by investigating other unsupervised techniques, and also implement SN on top of more GNN architectures to further demonstrate its applicability.

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
