# OpenReview forum: "Stabilized Self-training with Negative Sampling on Few-labeled Graph Data"
_ICLR.cc/2022/Conference — ICLR 2022 Submitted_

### Official Review · Reviewer_wfd3 · 2021-11-01

**Correctness:** 3
**Technical Novelty And Significance:** 3
**Empirical Novelty And Significance:** 3
**Recommendation:** 5
**Confidence:** 4

**Main Review:**

Strengths:
1. This paper proposes a stabilized self-training technique which the pseudo labels are update adaptively per epoch and the weights of different pseudo labels in the loss function are proposed to address the unstable issue of existing GNNs.
2. The author chooses negative samples from the nodes that are not directly connected and designs a new loss function to make positive and negative samples as different as possible.
3. The classification accuracy are reported when varying the number of labeled nodes per class in {1,3,5,10,20}, the results show the good performance compared with the baselines.

Weaknesses:
1. The proposed algorithm has some new ideas, such as negative samples, but the model of GCN is not changed.
2. The authors said that the unstable issue of existing GNNs can be addressed by the weights of different pseudo labels in the loss function, which named stabilizer in the paper. However, it has not been clearly proved by theoretical analysis or experiment.
3. The choices of negative samples and positive samples have some uncertainties. The authors should show the effect of the uncertainties.
4. Although the authors compared the proposed method with some unsupervised methods and supervised methods, some self-training or pseudo labeling methods for GCN or semi-supervised learning should be considered.
5. The performance of the method is not fully verified by experiment, for example, (1) the importance of stabilizer and the number of positive and negative samples are not reported. (2) the performance comparison between the proposed pseudo labels and existing pseudo labels.

Minor problem:
1. The figures can not match with the words in the body of article. For example, “The dashed lines are the ground-truth percentage of each class in the Cora dataset” or “the average number of predicted labels in percentage per class at each epoch”
2. There are many problems in the article, some of the description details remain in doubt, some of the statements are not exact. For example, (1) In the second paragraph of 3.2, “Fi representing the output probability vector of node vi ∈ V”, however, V is the node set. (2) In the last paragraph of 3.2, the author proposed “a pseudo label in previous epoch will be removed in the next epoch if its confidence becomes low.”, however, it hasn’t been depicted in the paper.


**Summary Of The Paper:**

This paper presents a self-training GCN framework. It refines the predicted results as pseudo labels and pseudo negative labels to train GCN, which is applicable to existing GCNs to stabilize the training process and enhance the training data.

**Summary Of The Review:**

The present paper presents an effective framework for semi-supervised node classification on few-labeled graph data. The paper is well organized and the motivation of this paper is clear. Although this paper has some innovation, but the model of GCN is not changed. Besides, some comparison experiments with pseudo labeling methods are lack.

---

### Official Review · Reviewer_S1e4 · 2021-11-02

**Correctness:** 3
**Technical Novelty And Significance:** 2
**Empirical Novelty And Significance:** Not applicable
**Recommendation:** 5
**Confidence:** 4

**Main Review:**

Strengths：
1. The proposed problem is interesting in the graph learning community. The problem is similar to the topic of few-shot learning on graphs, which usually assumes that only a few labeled nodes or graphs are available. Compared with the existing few-shot learning methods, the paper assumes all the unlabelled nodes are available at the training stage and addresses it with popular training technics such as self-training and negative sampling.
2. The paper is well organized with a clear problem definition and technical solution.
3. The proposed two strategies are easy to be reproduced.


Weaknesses:
1. Although the paper tries to address the challenging and exciting problem, i.e., semi-supervised node classification on graphs with an extreme case when only a small subset of nodes have class labels, my major concern is that the self-training and negative sampling are standard methods and have been discussed in many graph learning papers, as referenced by this paper. Therefore, without significant technical improvement, I think the technical contribution is limited.
2. I suggest the authors to make the Figure 1 more clear. It is not apparent to see the unstable training of GCN and DAGNN compared with the improved ones, due to the considerable overlap between 7 classes. For example, the variance of Class 3 in (a) seems to be smaller than the improved version shown in (c).
3. For the stabilized self-training, the ablation experiments need to be performed to show the differences between the proposed stabilized self-training and existing self-training.

4. Moreover, in implementing the stabilizer, a re-weighted pseudo-label loss (see Eq 5) is proposed, and the importance of each node is dependent on the number of their belonging classes. How to deal with the situation when a class has no node whose confidence is beyond the threshold? How to deal with the situation when a class dominates the most nodes on a graph? I suggest incorporating the neighbor information to design a more adaptive stabilizer.

5. The negative sampling method, “The negative samples of a positive sample vi are the nodes that are not directly connected to vi in the graph G”, may bring false negative samples because even two nodes are not directly connected, they still have the same class. How to deal with this problem?
6. I think the work can be better if the author considers problem 4 and 5.


**Summary Of The Paper:**

This paper considers the semi-supervised node classification on graph data with only a few node labels available. Under this extreme situation, the paper demonstrates the unstable performances of existing GNNs, such as GCN and DAGNN. To address the unstable problem, the paper proposes two strategies consisting of pseudo-labelling based self-training and negative-sampling based regularization. The experiments show that the two strategies work well on graph data with only a few node labels.

**Summary Of The Review:**

I think the main contribution is: the proposed problem is challenging and interesting, this paper considers the semi-supervised node classification on graph data with only a few node labels available. The main weakness is: the technical contribution is limited.

---

### Official Review · Reviewer_DW1F · 2021-11-02

**Correctness:** 2
**Technical Novelty And Significance:** 2
**Empirical Novelty And Significance:** Not applicable
**Recommendation:** 3
**Confidence:** 4

**Main Review:**

Strengths:
1. This paper addresses an important problem of semi-supervised node classification with very few training labels.
2. The few label problem is well motivated with a clear problem formulation.

Weakness:
1. The technical novelty of this paper is very limited. The proposed method is simply adapted from existing self-training and negative sampling approaches with incremental changes.
2. The reweighing scheme for pseudo labels is somewhat ad-hoc. The authors need to provide more convincing justifications why the reweighing approach can help mitigate the training unstability.
3. The proposed negative sampling method operates on labels rather than embeddings. However, When you choose negative samples that are not directly connected to a positive sample vi, there is no guarantee that the positive and negative sample would have different labels. How would you address this issue?
4. The proposed method is not compared with more recent baselines on self-training methods, for example,
Ziang Zhou, Shenzhong Zhang, and Zengfeng Huang. 2019.   Dynamic Self-training  Framework  for  Graph  Convolutional Networks.arXiv preprintarXiv:1910.02684(2019)
Ke Sun, Zhanxing Zhu, and Zhouchen Lin. 2020.  Multi-Stage Self-Supervised Learning for Graph Convolutional Networks. InAAAI.
5. The writing and clarity of the paper need be improved for better readability.

**Summary Of The Paper:**

This paper proposes a stabilized self-training with negative sampling method to improve GNN performance on the node classification task when only few labels are provided for training. The main idea of this paper is to use the reweighted self-training method and the negative sampling method to stabilize GNN.

**Summary Of The Review:**

The technical novelty and originality of the paper are limited. The proposed method is an adaptation of the existing self-training and negative sampling strategies. This paper lacks comparisons with more recent self-training baselines to prove the effectiveness of the proposed method.

---

### Official Review · Reviewer_XgQF · 2021-11-02

**Correctness:** 2
**Technical Novelty And Significance:** 1
**Empirical Novelty And Significance:** 1
**Recommendation:** 3
**Confidence:** 5

**Main Review:**

Strength

1 - Propose a new method for node classification with few-shot labels.

2 - Presentation is clear for me.

Weakness

1 - The novelty of this work is limited.

2 - Miss important related works.

3 - Experiments should be improved.

Detailed Review

The novelty of proposed method is limited for me. The major contribution: data augmentation strategy (pseudo label) and negative sampling regularization, is simple and intuitively, depending on the threshold value. There are many works proposing better data augmentation strategies or pseudo label generation, such as:

Graph contrastive learning with augmentations, NeurIPS 2020

Big Self-Supervised Models are Strong Semi-Supervised Learners, NeurIPS 2020

More importantly, this paper lacks discussion and comparison of important related works. The authors seem to be not aware of graph few-shot learning works. There are many studies working on graph learning with few-shot labels, such as followings:

Graph meta learning via local subgraphs, NeurIPS 2020

Graph few-shot learning via knowledge transfer, AAAI 2020

Meta-gnn: On few-shot node classification in graph meta-learning, CIKM 2019

Node classification on graphs with few-shot novel labels via meta transformed network embedding, NeurIPS 2020

**Summary Of The Paper:**

This paper proposes a self-training method with negative sampling for node classification on few-labeled graph data. The proposed method applies data augmentation (i.e., pseudo label) and negative sampling regularization to augment node classification model. Experiments are conducted to show that the proposed method outperforms some baseline methods.

**Summary Of The Review:**

The novelty of this work is limited. It lacks important related work (graph few-shot learning) discussion and comparison. In my opinion, this work is not suitable for ICLR.

---

### Official Review · Reviewer_s9Vo · 2021-11-03

**Correctness:** 2
**Technical Novelty And Significance:** 1
**Empirical Novelty And Significance:** Not applicable
**Recommendation:** 1
**Confidence:** 4

**Main Review:**

Both the problem setting and technical contribution of this paper is problematic. First of all, the proposed method is basically a bootstrapping-style self-training method, it needs to compare with other self-training methods instead claiming it can improve the performance. The comparison in the experiment is unfair, as proposed method utilizes way more training seeds. Second, the **unstable** phenomenon is not defined in a mathematical way. It's widely known that few training seeds leads to larger variance. The terminology here looks like proposing a new measurement without any mathematical support.

1. **Lack of proper baselines**
As a self-training method, author should compare more baselines that considers additional training inputs. Also, experiments on larger benchmarks are necessary. It's a common sense that an algorithm can work well on Cora/Citeseer/PubMed because the network is simple and small.

2. **Limited novelty**
The threshold-based detector has been widely used in out-of-distribution detection [1]. The proposed method is neither novel nor insightful.



[1] Ren, Jie, et al. "Likelihood Ratios for Out-of-Distribution Detection." Advances in Neural Information Processing Systems 32 (2019): 14707-14718.

**Summary Of The Paper:**

This paper discusses *unstable* training procedure of graph neural network when training data is extremely limited. The author proposes use self-training and negative sampling to mitigate this issue. Specifically, it uses high-confidence prediction of the model as training seeds and control the importance by population. The result shows improvement on several node classification dataset.

**Summary Of The Review:**

Overall, I recommend a rejection to this paper. The problem it studied is a common challenge (amount of training data vs. variance) in semi-supervised learning, while experiments are not conducted properly to support the effectiveness of it against other similar approaches. Moreover, the technical writing in the paper contains a lot of hand-waving example instead of sound mathematical justifications.

---

### Decision · Program_Chairs · 2022-01-20

**Decision:**

Reject

**Comment:**

The paper studies stability issues of GNN training when data are limited. The key contribution of this work is to use reweighted self-training and negative sampling to stabilize GNN. Multiple reviewers raised major concerns on the technical novelty, experimental setup, comparison, and results. No response was provided during discussion. I recommend this submission be rejected.